# The Use of the TARGET Antibiotic Checklist to Support Antimicrobial Stewardship in England’s Community Pharmacies

**DOI:** 10.3390/antibiotics12040647

**Published:** 2023-03-24

**Authors:** Sejal Parekh, Catherine V. Hayes, Jill Loader, Diane Ashiru-Oredope, Kieran Hand, Gemma Hicks, Donna Lecky

**Affiliations:** 1Primary Care Group, Primary, Community and Personalised Care Directorate, NHS England, London SE1 8UG, UK; 2HCAI, Fungal, AMR, AMU & Sepsis Division, UK Health Security Agency, London SW1P 3HX, UK; 3AMR Programme, Medical Directorate, NHS England, London SE1 8UG, UK

**Keywords:** community pharmacy, pharmacy quality scheme, antimicrobial stewardship, primary care, medication safety, antimicrobial resistance, incentivisation

## Abstract

Antimicrobial Stewardship (AMS) requires effective teamwork between healthcare professionals, with patients receiving consistent messages from all healthcare professionals on the appropriate antimicrobial use. Patient education may reduce patients’ expectations to receive antibiotics for self-limiting conditions and reduce the pressure on primary care clinicians to prescribe antibiotics. The TARGET Antibiotic Checklist is part of the national AMS resources for primary care and aims to support interaction between community pharmacy teams and patients prescribed antibiotics. The Checklist, completed by the pharmacy staff with patients, invites patients to report on their infection, risk factors, allergies, and knowledge of antibiotics. The TARGET antibiotic checklist was part of the AMS criteria of England’s Pharmacy Quality Scheme for patients presenting with an antibiotic prescription from September 2021 to May 2022. A total of 9950 community pharmacies claimed for the AMS criteria and 8374 of these collectively submitted data from 213,105 TARGET Antibiotic Checklists. In total, 69,861 patient information leaflets were provided to patients to aid in the knowledge about their condition and treatment. 62,544 (30%) checklists were completed for patients with an RTI; 43,093 (21%) for UTI; and 30,764 (15%) for tooth/dental infections. An additional 16,625 (8%) influenza vaccinations were delivered by community pharmacies prompted by discussions whilst using the antibiotic checklist. Community pharmacy teams promoted AMS using the TARGET Antibiotic Checklist, providing indication-specific education and positively impacting the uptake of influenza vaccinations.

## 1. Introduction

Antimicrobial resistance (AMR) is defined as the loss of antimicrobial effectiveness against microorganisms and, although it develops naturally, this process is accelerated by the inappropriate or incorrect use of antimicrobials [1]. AMR is a global problem and was associated with 4.95 million deaths globally in 2019 [2]. The rising prevalence of AMR is likely to cause the increased suffering and potential deaths attributable to it, leading to increased socio-economic costs associated with treating ill health in humans [3].

Antimicrobial stewardship (AMS) refers to an organisational or healthcare system-wide approach to promoting and monitoring the judicious use of antimicrobials to preserve their future effectiveness [4]. The UK AMR National Action Plan sets out the 2019–2024 commitments to tackle AMR within and beyond our own borders and has three overarching aims:reducing need for, and unintentional exposure to, antimicrobials;optimising use of antimicrobials;investing in innovation, supply and access [3].

Although a reduction in the prescribing of antimicrobials in primary care has been observed in England, the majority of antibiotics are still prescribed in this sector [5]. In 2020, 80% of antibiotic prescribing occurred in primary care in England (72% in general practice; 4% dental; 4% other community settings) [5].

In the UK, community pharmacies are recognised as a first port of call for healthcare advice and services because of the convenient access to a health care professional for managing minor ailments and self-limiting illnesses [6]. They have successfully been able to manage a range of clinical services and public health initiatives for patients who may otherwise seek assistance from their general practitioner (GP) [6]. 

The UK AMR National Action Plan has recognised that one of the most successful initiatives used to embed AMS into practice has been the use of the TARGET Antibiotic Toolkit jointly produced by the UK Health Security Agency (UKHSA) and the Royal College of General Practitioners (RCGP) [7]. The toolkit provides a range of resources to support health professionals facilitate AMS whilst educating patients on the appropriate use of antibiotics [7]. The TARGET Antibiotic Checklist is designed to be completed by pharmacy staff with patients when they present with an antibiotic prescription. It asks patients to report their infection, risk factors, allergies, and knowledge of antibiotics; this information allows pharmacy teams to tailor their responses and counsel patients specific to their infection and their knowledge [8]. The TARGET Antibiotic Checklist provides a reference for structuring conversations about antibiotics with patients as well as tailoring counselling and addressing gaps in knowledge [9,10].

The Pharmacy Quality Scheme (PQS) is part of the community pharmacy contractual framework (CPCF) for England [11]. The scheme incentivises initiatives that focus on three overarching objectives: clinical effectiveness, patient safety, and patient experience, as well as the delivery of the National Health Service (NHS) long term plan [11,12]. The scheme consists of gateway criteria that allow community pharmacies to be eligible for a PQS payment. Each criterion is grouped to a domain and they must complete all criteria within this domain to receive payment via allocated points. 

Since 2020, the scheme has incentivised AMS activities in community pharmacies [13]. These included asking community pharmacy teams to complete the UKHSA: AMS for Community Pharmacy e-Learning programme (hosted on Health Education England e- learning for health platform); developing an action plan as to how they would apply the learning and tackle AMR in their pharmacy; a pledge to become an Antibiotic Guardian; and increasing awareness of local antibiotic formulary [14,15]. Each year, the PQS aims to build upon previous initiatives to maximise the learning and development in bite-size chunks as well as embed initiatives into day-to-day practice. In 2021/22, the PQS incentivised community pharmacy teams to use the TARGET Antibiotic Checklist for patients presenting with an antibiotic prescription [9,16]. The aim of this study was to review and evaluate the findings from the implementation of the TARGET Antibiotic Checklist for all community pharmacies in England as part of the 2021–22 PQS [14]. The TARGET Antibiotic Checklist was first piloted as part of the pharmacy AMS intervention (PAMSI) [9]. The PAMSI intervention, incorporating the use of TARGET Antibiotic Checklist, was further evaluated in community pharmacies [10]. This paper reports on the evaluation of the national scale up of the TARGET Antibiotic Checklist for use in community pharmacies.

Evaluation of the clinical findings and implementation barriers are reported separately [17].

## 2. Materials and Methods

### 2.1. Study Design

This is a service evaluation of the scaling of the use of the TARGET Antibiotic Checklist in community pharmacies via the national incentive scheme. In this structured observational study, researchers assessed the outcomes of how community pharmacists implemented the criteria within the AMS domain of the PQS criteria. Data were collected as a part of England’s Pharmacy Quality Scheme (PQS) 2021–22 [14]. The scheme criteria were developed by NHS England in collaboration with UKHSA and the Pharmaceutical Services Negotiating Committee (PSNC). The specific requirement of the TARGET Antibiotic Checklist was part of the prevention domain, building on previous schemes’ AMS activities from PQS Year 2020–21 and 2021–22. 

### 2.2. Setting and Participants

The PQS is a voluntary scheme and all community pharmacies in England providing NHS services are eligible to participate [18]. Incentive payments can be accessed once certain gateway criteria are met. In March 2022, there were 11,232 registered community pharmacies in England [19]. Participating community pharmacies collected data from patients presenting with an antibiotic prescription. Any patient presenting with an antibiotic prescription was eligible for inclusion. The information requirements and guidance about the scheme were communicated by the Department of Health and Social Care (DHSC) via the Drug Tariff determination and NHS England PQS Guidance [13,20]. Further supporting information for contractors was provided by the Pharmaceutical Service Negotiating Committee (PSNC) [18]. In March 2022, UKHSA distributed 2 × A4 laminated TARGET Antibiotic Checklists to every community pharmacy on the NHS England pharmaceutical list to support the implementation of the tool in community pharmacies. 

### 2.3. Data Collection

Pharmacy teams were asked to use the TARGET Antibiotic Checklist within their current AMS practice for four weeks with a minimum of 25 patients, or up to eight weeks if the minimum number of patients was not achieved within four weeks [8]. Contractors could declare at eight weeks if they had not reached the minimum sample size. (Pharmacy teams could be comprised of pharmacists, pharmacy technicians, trainee pharmacists, trainee pharmacy technicians, dispensary staff, and medicines counter assistants). Pharmacy teams were required to submit the data via the UKHSA online Snap Survey portal on the date of their declaration and no later than 31 May 2022. Pharmacy teams could complete the criteria from the launch of the scheme on 1 September 2021 to its closure on 31 May 2022 to claim the incentive. 

The online Snap Survey (Snap 11 Professional) portal digitalised the questions from the TARGET Antibiotic Checklist as well as included demographic information such as the pharmacy’s Organisation Data Service (ODS) code and address [21]. A single form submission was required per prescription. The online data collection tool was piloted with 30 community pharmacy staff. Nine provided feedback to improve the tool prior to its finalisation. 

### 2.4. Data Analysis

The data was analysed using Microsoft Excel for descriptive analysis. As the tool allowed multiple answers to be ticked simultaneously, the research team agreed that reports of two or more concurrent infections were likely to be erroneous data entries. Therefore, after the initial analysis, the data were cleaned and re-analysed to only include patients reported to have been prescribed up to two antibiotics for a maximum of two indications at that specific point in time. Further ambiguous data were removed along the way—e.g., if both options (yes and no) were ticked for subsequent questions, these data were removed from the analysis. From the 213,105 checklists reported, 208,858 (98%) were included in the final data analysis. 

### 2.5. Ethics

As this study is a service evaluation, NHS ethical approval is not required [22]. Institutional ethical approval was not required as confirmed through the UKHSA Research Ethics and Governance Group and the NHS Health Research Authority Decision tool [22]. Data was collected as a part of provider assurance to confirm a community pharmacy has completed the requirement for the PQS. No patient identifiable data was submitted in the data collection. The data collection tool was hosted on a secure UKHSA Snap Survey platform licensed as Snap 11 Professional. All data collection tools provided information on the purpose of collecting the data and how it would be used. All files were handled in accordance with the Data Protection Act 2018 and the General Data Protection Regulations (GDPR). 

## 3. Results

### 3.1. PQS Uptake

The total number of community pharmacy contractors who claimed to be for the PQS prevention domain was 9950 (of which 8374 submitted data via the online portal) out of the 11,232 community pharmacies listed on the pharmaceutical list in March 2022. Pharmacy contractors collectively submitted data for 213,105 TARGET Antibiotic Checklists. 

### 3.2. Antibiotic Use Background Information

A total of 163,446 (78%) of the submitted TARGET Antibiotic Checklists were completed for patients who were collecting their own antibiotics, and the remainder on behalf of patients for whom a representative was collecting their antibiotics. From the submitted checklists, 107,360 (51%) patients reported taking other medications alongside their antibiotics (Table 1) and 165,383 (79%) of patients reported they were not allergic to any antibiotics.

### 3.3. Patient Infections and Antibiotics Dispensed

The most common indication for which a checklist was completed (and, as such, an antibiotic prescribed) was for a respiratory tract infection (RTI), representing 62,664 (29%) of submissions, followed by a urinary tract infection (UTI) with 43,093 (20%) submissions, and tooth infections with 31,198 (14%) submissions (see Figure 1). 

The most frequently prescribed antibiotics were amoxicillin, accounting for 73,921 (35%) submissions, followed by nitrofurantoin (30,459, 14%), flucloxacillin (24,093, 11%), doxycycline (19,420, 9.2%), and metronidazole (13,720, 6.5%) (see Figure 2). Where two antibiotics were prescribed, the most common antibiotic combination was amoxicillin and metronidazole, which accounted for 1754 (0.8%) submissions.

### 3.4. Provision of Information to Patients

Pharmacy teams reported providing 69,861 patient information leaflets with patients’ dispensed antibiotics to aid with patients’ knowledge about their condition and treatment (See Figure 3). The majority of leaflets provided corresponded to the indication given by the patients. 

The TARGET Antibiotic Checklist aims to facilitate targeted information sharing between patient and pharmacist to address concerns and give appropriate counselling. Patients most frequently selected that they did not understand the following side effects of their antibiotics (34,560, 16.6%), how to take their antibiotics with food (32,247, 15.5%), and how long their symptoms would last (32,247, 15.5%). There was selection for information about why they must take their antibiotics according to healthcare professional instruction (8809, 4.2%), why they must never share or keep their antibiotics for later use (12,030, 5.8%), and knowing when to seek further help with their infection (14,656, 7.0%) (See Figure 4).

In the patient section of the TARGET Antibiotic Checklist, the patient reported in 36,097 (17%) of cases that they had previously received the same antibiotic in the last three months. Pharmacy staff discussed antibiotic resistance with patients for 87,975 (42%) checklists (See Table 2).

### 3.5. Influenza Vaccinations

Overall, 97,463 (47%) of the submitted checklists confirmed that the patient had already received the influenza vaccination, of which 53,592 (25%) were eligible for a free NHS influenza vaccination (See Table 3)). (All adult patients can receive an influenza vaccination but the most at risk are entitled to a free NHS vaccination) [23]. Out of the eligible population who were entitled to free NHS influenza vaccines (53,592), 39,960 (76%) of the Antibiotic Checklists indicated that the patient had already received it; for the subgroup of patients aged 65 years and above, 36,712 (77%) patients confirmed they had already received it (See Table 3). As a result of the interventions from the use of the Antibiotic Checklist, 16,625 (8% of all submitted Antibiotic Checklists) additional influenza vaccinations were delivered by community pharmacies, with 5019 (30%) being for patients eligible for free NHS influenza vaccinations and 11,606 (70%) for patients not eligible for a free NHS vaccination, according to information provided (See Table 3). 

## 4. Discussion

### 4.1. Uptake of the TARGET Antibiotic Checklist 

The initiative proved successful in terms of both uptake and sharing information with patients. In total, 89% of all community pharmacies in England participated in the Pharmacy Quality Scheme AMS criteria in 2021–22, which is consistent with previous pharmacy quality schemes [24]. As expected for primary care, RTI and UTI were the most common types of infections that patients presented in community pharmacies with prescriptions for [25].

Research has demonstrated that the TARGET resources facilitate pharmacist–patient communication [9,26]. In this study, community pharmacy teams issued a total of 69,861 patient information leaflets to patients alongside their antibiotic prescription, discussed antibiotic resistance with the patient/carer with 87,975 (42%) checklists, and administered a further 16,625 (8% of all checklists) influenza vaccinations. In addition, the Antibiotic Checklist facilitated numerous conversations and tailored counselling on how patients should take their prescribed antibiotic.

Community pharmacies have been administering influenza vaccinations as an advanced service since 2015 [23]. As well as being a key element of NHS resilience and aimed at reducing winter pressures, vaccination programmes are part of an alternative approach to tackling AMR. Vaccinations promote herd immunity by preventing patients from contracting influenza and/or reducing the risk of severe illness [23]. This in turn reduces secondary bacterial infections and the associated patient exposure to antibiotics and antivirals, which contributes to slowing the rate of the development of antibiotic resistance [3,23]. 

The initiative highlighted the risks of relying on patient recall alone when establishing medication history as only 17% (36,097) of patients confirmed they had taken the same antibiotic in the past 3 months when the rate indicated from the pharmacy staff counselling records was 42% (87,975). Whilst it can be argued that most patients are unlikely to have medical training to remember the exact name of their antibiotic and then make the association that it was the same antibiotic, pharmacy teams should take care if relying on patient recall to provide information and consider confirming with dispensing system records when appropriate. 

Community pharmacy teams provided advice to 59,168 (28%) patients collecting an antibiotic as to why unused antibiotics need to be returned to a community pharmacy for disposal. It is unclear the extent of patient comprehension, i.e., whether patients understood that if antibiotics are not used responsibly, they can lead to dangerous side effects, a delay to an accurate diagnosis, contribute to antimicrobial resistance (AMR), and/or whether inappropriate disposal of unused antibiotics also damages the environment, impacting water quality and wildlife [27]. Regardless, the results suggest a lack of patient awareness that could be improved upon. It is therefore important for community pharmacies to build on the awareness of their local patient population, explaining why it is necessary for unused antibiotics to be returned to a pharmacy for destruction. Counselling on returning unused antibiotics, often referred to as an “Antibiotic Amnesty” has been proposed for inclusion as part of the PQS 2023–24 [27]. 

### 4.2. The Role of Community Pharmacy in Managing Infections

Many patients attend community pharmacies for the treatment and management of lower acuity health issues or minor ailments. In fact, the development of the Community Pharmacist Consultation Service (CPCS) has diverted many patients from NHS 111 (telephone triage), out-of-hours services, and from GP practices to community pharmacies for management of these minor ailments [28]. The top three symptom groups for CPCS referral to date have been influenza/colds, cough, and sore throats as per unpublished data held by NHS England.

Research has shown that individuals prescribed an antibiotic in primary care for a RTI or UTI are more likely to harbour bacteria resistant to that antibiotic [29]. The effect is greatest in the month immediately after treatment but may persist for up to 12 months [29]. This effect not only increases the population carriage of organisms resistant to first line antibiotics, but also creates the conditions for the increased use of second line antibiotics in the community. Repeated courses of the same antibiotic can compound this effect. Counselling on antibiotic use, as well as recommending switching between antibiotics for repeated UTI infections (as per NICE guidance), is expected to reduce the risk of treatment failure and AMR selection pressure, as well as contribute to managing these patients appropriately [30]. Every time a patient attends a GP practice with RTI and UTI symptoms, they have 50–100% chance of receiving a course of antibiotics [29]. Pharmacists can discourage unnecessary visits to physicians and provide self-care advice when appropriate [31]. With this in mind, it is crucial that the community pharmacy team know how to manage all patients with such symptoms (be it walk-in patients or those presenting with a prescription) and when to escalate or reassure that this is the normal and self-limiting trajectory of the disease [7]. To support the pharmacy team, the use of TARGET “treating your infection” leaflets was incentivised as part of PQS 2022–23 to manage walk-in patients whilst the TARGET Antibiotic Checklist supports pharmacy staff to empower patients to self-care and provides counselling about their antibiotic prescriptions. These initiatives may help to reduce future antibiotic-seeking behaviour.

Continuing the initiative alongside evaluation will contribute to reinforcing AMS principles through the continued practice of using the TARGET Antibiotic Checklist for patients prescribed antibiotics as well as promoting sustained change for its use. The success of the TARGET Antibiotic Checklist in the community pharmacy setting has led to the development of a new community pharmacy section on the TARGET Antibiotic Toolkit website: Resources for community pharmacy setting to aid pharmacy teams in promoting AMS [32]. The dedicated TARGET tools and resources are aimed specifically at community pharmacies to enable pharmacy staff to promote AMS and practical behaviours in a streamlined way. 

### 4.3. Strengths and Limitations

Much like the study “Evaluation of the England Community Pharmacy Quality Scheme (2018–2019 and 2019–2020) in reducing harm from Non-Steroidal Anti-inflammatory Drugs in older patients” where the PQS was used to promote medicines safety, the dataset has a high ecological validity as data have been collected from every day practice, giving a snapshot of community pharmacy AMS activities [33]. 

Over 200,000 data entries were collected as part of this PQS criteria from all over England, one of the largest studies conducted on community pharmacies to date [17]. This quantitative analysis highlighted the role that community pharmacies can play in promoting AMS activities amongst their patient population and how, collectively, this will contribute to reducing AMR [34]. The AMS initiatives in PQS allowed the majority of community pharmacies to receive the same training and education simultaneously to promote consistent practices with their local population and collectively across England. Practically, community pharmacy teams are now familiar with TARGET tools and resources available to them to discuss AMS with patients and to address individual patient’s concerns and queries confidently. Overall, the collective AMS PQS initiatives have set a sound foundation of community pharmacies promoting AMS and contributing to curbing the AMR. In addition, this work has primed community pharmacies for future developments.

A study limitation is that the data were self-reported by community pharmacy teams without independent validation, so both inconsistent or inaccurate reporting cannot be ruled out. Data cleaning helped to remove duplicate entries, erroneous entries, and data absence. It could also be argued that pharmacy teams may behave differently to usual practice when taking part in an incentivised scheme such as the PQS, with potential temporary changes in behaviours; however, more checklists were submitted than were required for the purposes of the PQS and there is a known continuation of resource use post intervention period [17]. Additionally, it is challenging to estimate the longer-term impact and sustainability of this PQS scheme in the absence of national standards/framework for community pharmacies and antimicrobial stewardship to tackle antimicrobial resistance.

Pharmacy teams were allowed to check multiple response options simultaneously on the data collection tool as some patients could have multiple infections. This, however, may have led to erroneous data entry, for example, checklist data suggesting nitrofurantoin and trimethoprim, antibiotic items typically prescribed for UTIs, being prescribed for chest infections. However, after data cleaning, 208,858 (98%) out of the 213,105 checklists were included in the analysis, suggesting limited human error. The upcoming PQS will use the findings from this study to refine the data collection process and data collection tool.

All major AMR control and AMS strategies recommend ongoing education for patients, children, the public, and relevant healthcare professionals (e.g., primary-care physicians, pharmacists, and related students) regarding unique features of bacterial infections and antibiotics, prudent antibiotic prescribing as a positive construct, and personal hygiene (e.g., handwashing) [35]. AMR can only be effectively reduced by the ongoing and rigorous efforts from all healthcare professionals and the general population as a whole [35]. 

## 5. Implications and Conclusions

The majority of antimicrobial drugs are prescribed and dispensed in primary care, hence the structured and ongoing education on AMS for primary care healthcare professionals and patients is paramount [36]. Community pharmacists recognise the need to implement AMS activities in their day-to-day practice [26]. They are often the first port of call and are in a unique position to provide triage for common primary care indications and to lower the burden of patients at general practitioners’ surgeries. However, the setting makes it difficult to have quick contact with prescribers, as well as currently being unable to change a prescription without contacting the prescriber.

Previous research has demonstrated the value of using the TARGET Antibiotic Checklist to support community pharmacy teams who play a crucial role in AMS by using targeted patient education and addressing patients’ concerns about antibiotic use [9,10]. This study demonstrates how this tool can be implemented at the national level, which may further reduce patient expectation to receive antibiotics for self-limiting infections and reduce pressure on general practitioners to prescribe antibiotics on the patients’ demand. 

Whilst engaging in conversations about their antibiotic prescriptions, community pharmacy teams have provided additional indication-specific education materials to reinforce their counselling and increased the influenza vaccinations uptake. In light of this, the initiative has been renewed as part of the NHS England Community Pharmacy Quality Scheme 2023–24, which will support embedding the use of the TARGET resources into day-to-day practice and reinforcing AMS behaviours within community pharmacies [37]. 

AMR is an issue of international concern. This intervention could be adapted for international use as it would be suitable for use for all patients (globally) presenting with an antibiotic prescription. The TARGET Antibiotic Checklist is available digitally and translated in multiple languages [8]. In addition, the managing common infection (self-care) leaflets are also available digitally and translated in multiple languages [38]. These resources can be used for patients accessing services remotely as well as internationally.

This study highlights the vital role that community pharmacy teams have in promoting AMS initiatives, including educating the public to ensure the safe and appropriate use of antibiotics. 

## Figures and Tables

**Figure 1 antibiotics-12-00647-f001:**
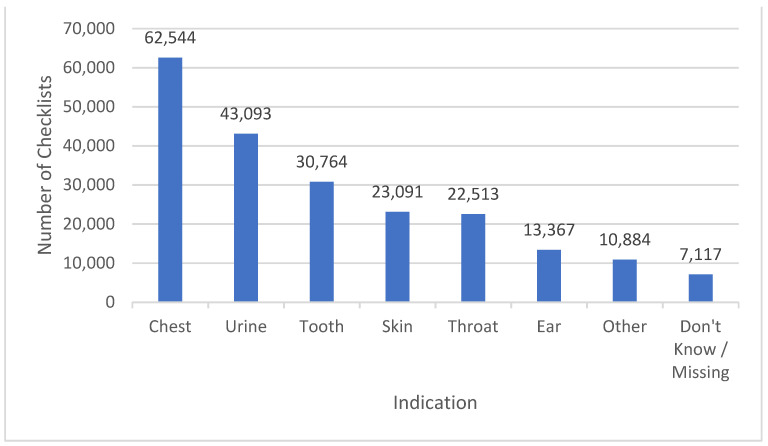
The frequency of infection type reported by patients collecting antibiotic prescriptions. Note: Some checklists had two conditions ticked.

**Figure 2 antibiotics-12-00647-f002:**
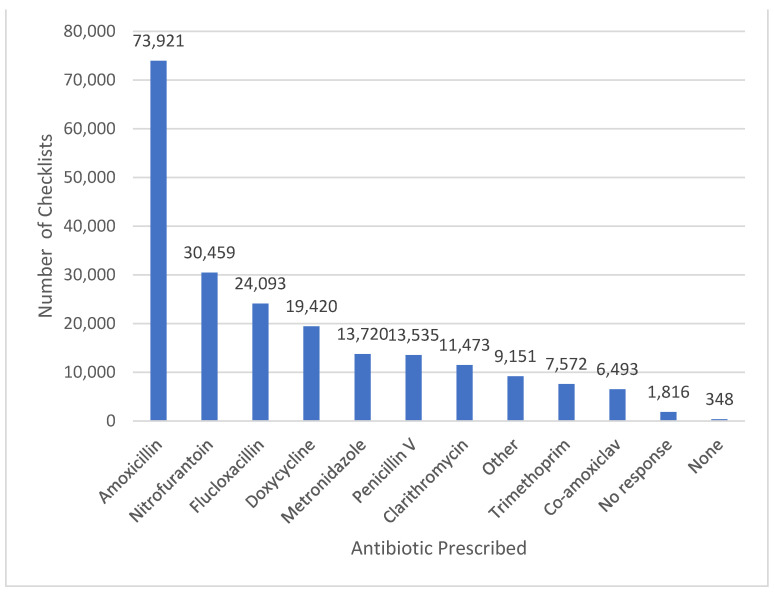
The frequency of antibiotics prescribed to patients for whom the TARGET Antibiotic Checklist was completed.

**Figure 3 antibiotics-12-00647-f003:**
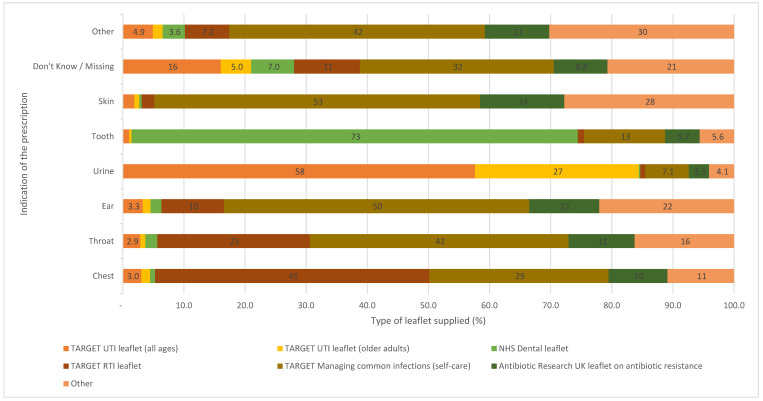
Percentage and type of supporting information leaflets provided to patients by antibiotic checklist reported indication.

**Figure 4 antibiotics-12-00647-f004:**
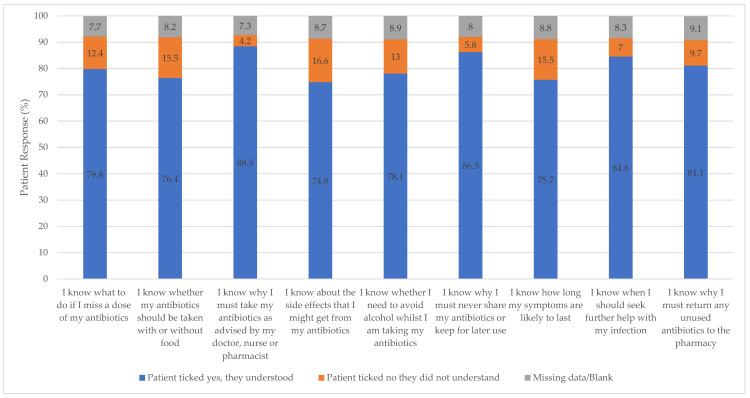
Patient self-reported knowledge of specific information relevant to infection and safe and effective antibiotic treatment.

**Table 1 antibiotics-12-00647-t001:** Findings from Patient response section of the Antibiotic Checklist.

	Were the Antibiotics for the Individual Collecting? %	Was the Patient Taking Any Other Medicines? %	Was the Patient Allergic to Any Antibiotics? %
Yes	163,446 (78)	107,360 (51)	25,574 (12)
No	42,123 (20)	90,764 (43)	165,383 (79)
Do not know/missing	1954 (0.9)	8307 (4.0)	14,867 (7.1)
No Response	1335 (0.6)	2427 (1.2)	3034 (1.5)
Total	208,858 (100)	208,858 (100)	208,858 (100)

**Table 2 antibiotics-12-00647-t002:** Responses from both patient and pharmacist of using the same antibiotic in the last 3 months (n = number of checklists).

Response	Patient-Reported Prescription of the Same Antibiotic(s) within Previous 3 Months (%)	Pharmacy Staff Conversation with Patient about Resistance Due to Prescription of the Same Antibiotic(s) within the Previous 3 Months (%)
Yes	36,097 (17)	87,975 (42)
No	156,836 (75)	35,168 (17)
Not applicable	0	67,210 (32)
Do not know/missing	13,203 (6.3)	9320 (4.5)
No response	2722 (1.3)	7490 (3.6)
Total checklists	208,858	207,163 *

* This was filtered to exclude checklists where multiple answers were ticked, i.e., if yes and no were ticked.

**Table 3 antibiotics-12-00647-t003:** The number of TARGET Antibiotic Checklists indicating patients were eligible to receive a free influenza vaccination.

Influenza Vaccination Eligibility Category	No of Patients (% of Total)	No of Influenza Vaccines Already Received (% by Eligibility Criterion)	Gave the Patient an Influenza Vaccine on Site (% of Those Who Had Not Yet Received Vaccine)
Over 65 years	47,622 (23)	36,712 (77)	4431 (41)
Problem with kidney function *	2968 (1.4)	1648 (56)	265 (20)
Problem with liver function *	740 (0.62)	388 (52)	85 (24)
Pregnant *	2262 (1.1)	1212 (54)	238 (23)
Total	52,592 (26)	39,960 (76)	5019 (36)

* Under 65.

## Data Availability

The data presented in this study are available on reasonable request from the corresponding authors.

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
