# Peer review of "The Use of the TARGET Antibiotic Checklist to Support Antimicrobial Stewardship in England’s Community Pharmacies"

_antibiotics, 2023, doi:10.3390/antibiotics12040647_

Round 1
Reviewer 1 Report
The paper titled "The Use of the TARGET Antibiotic Checklist to support Antimicrobial Stewardship, in England’s community pharmacies through the Pharmacy Quality Scheme" presents an interesting study conducted to evaluate the effectiveness of the TARGET Antibiotic Checklist in promoting appropriate antibiotic use in community pharmacies. While it is promising to see that the checklist implementation helped patients to involve more with pharmacists to enhance knowledge about how to use antibiotics and even to support vaccination coverage, I think there are several areas where the paper could be improved to make it more scientifically robust.
Firstly, Although the authors have presented descriptive results of what the checklists contain, however as a scientific paper, I would like to know more about what the scientific question that the study aimed to answer as a research article. For instance, it is not clear how the checklist was developed, and whether there are any valid outcomes that can be used to measure the effect of the intervention. Moreover, on what level did the checklist policy improve the cognitive level of the patients? Furthermore, it would be helpful to understand if the intervention is repeatable in other settings, and how sustainable and scalable the intervention is.
Additionally, if the paper was focused more on the implementation of the policy, then it can be addressed in more detail. What are the barriers during implementation, and is the policy implemented with the same quality across different pharmacies? Moreover, it would be interesting to know the cost-effectiveness of the intervention.
While the paper provides a good report of the descriptive results of the checklist, it lacks the scientific rigor required for a scientific paper. There are many questions that could be raised and answered if the authors wanted to evaluate the policy more rigorously. I recommend that the authors to address these points and strengthen the scientific quality of the study.
Author Response
AMS Manuscript feedback response
Reviewer 1
The paper titled "The Use of the TARGET Antibiotic Checklist to support Antimicrobial Stewardship, in England’s community pharmacies through the Pharmacy Quality Scheme" presents an interesting study conducted to evaluate the effectiveness of the TARGET Antibiotic Checklist in promoting appropriate antibiotic use in community pharmacies. While it is promising to see that the checklist implementation helped patients to involve more with pharmacists to enhance knowledge about how to use antibiotics and even to support vaccination coverage, I think there are several areas where the paper could be improved to make it more scientifically robust.
Thank you for your feedback. Please see responses to your queries below.
Firstly, Although the authors have presented descriptive results of what the checklists contain, however as a scientific paper, I would like to know more about what the scientific question that the study aimed to answer as a research article. For instance, it is not clear how the checklist was developed, and whether there are any valid outcomes that can be used to measure the effect of the intervention.
These are also now reference in this manuscript
Moreover, on what level did the checklist policy improve the cognitive level of the patients?
The detail of this is part of the referenced papers and the initial evaluation of the use of TARGET and also addressed in the accompanied paper submitted to Antibiotics: Hayes, C., et al., The National implementation of a Community Pharmacy Antimicrobial Stewardship Intervention (PAMSI) through the English Pharmacy Quality Scheme 2020 to 2022.
Furthermore, it would be helpful to understand if the intervention is repeatable in other settings, and how sustainable and scalable the intervention is.
The antibiotic checklist has been developed to be used when patients present with a prescription for an antibiotic and therefore can be used for any patients presenting a prescription. In the majority of cases, we would assume that this would be at the pharmacy/dispensary where a patient would receive their medication.
Additionally, if the paper was focused more on the implementation of the policy, then it can be addressed in more detail. What are the barriers during implementation, and is the policy implemented with the same quality across different pharmacies? Moreover, it would be interesting to know the cost-effectiveness of the intervention.
While the paper provides a good report of the descriptive results of the checklist, it lacks the scientific rigor required for a scientific paper.
This is part of a service evaluation, reporting data from a national scheme. However, we will be able to do more statistical evaluation once we have the subsequent years' data to compare.
There are many questions that could be raised and answered if the authors wanted to evaluate the policy more rigorously. I recommend that the authors to address these points and strengthen the scientific quality of the study.
Thank you – I hope we have answered your queries.

Reviewer 2 Report
The article entitled The Use of the TARGET Antibiotic Checklist to support Anti- 2
microbial Stewardship, in England’s community pharmacies 3 through the Pharmacy Quality Scheme and aimed to support interaction between community pharmacy teams with patients prescribed antibiotics. This study concluded that Community Pharmacy teams promoted AMS using the TARGET Antibiotic Checklist, providing indication- specific education, and positively impacting on uptake of influenza vaccinations.
There are some MAJOR and fundamental check that author should provide answers:
1. How are formed “pharmacy teams” in study, please describe. What that means?
2. Please add a calculation for sample size.
3. How is performed initial analysis of data? Please specify.
4. The most important question is about ethical concerns. Why authors mentioned that there is no need for approval. In any case, with or without identifying the patients, authors must obtain the ethical consideration or consent at least (Please see declaration of Helsinki). This sentence : “Institutional ethical approval was also not required as patient identifying data were not collected.” Not true.
5. Is the SnapSurvey licensed? Please add a number of license.
6. From the results exclude the number of No response
7. Table 1 should be renamed, General findings are not appropriate. Please be more specific.
8. All Figures must be labels, both X and Y axis must be with the explanation. Also, what present the bars in Figures, percents?? Please add in Figures.
9. How you were sure in these data since there is type of self-reporting with no any controlling? How do you reduced the bias?
10. Do you have any previous pilot study to validate method tool?If you not, this results remain questionable as well as conclusions….
11. In section Conclusion erase the implications, since they could be based e just on thinking not on the results. Only conclusions are based on results.
12. The article is confused and very hard to read. There are a lot of abbreviations which reading keeps the more difficult, please reduce it.
13. Informed consent “All participants consented to participate and were aware of the study aims.” I am not sure about this. Please erase and obtain consent from the Institution to collect and use this data.
Author Response
AMS Manuscript feedback response
Reviewer 2
The article entitled The Use of the TARGET Antibiotic Checklist to support Antimicrobial Stewardship, in England’s community pharmacies through the Pharmacy Quality Scheme and aimed to support interaction between community pharmacy teams with patients prescribed antibiotics. This study concluded that Community Pharmacy teams promoted AMS using the TARGET Antibiotic Checklist, providing indication- specific education, and positively impacting on uptake of influenza vaccinations.
|
Thank you for your feedback – please see response below to your queries.
There are some MAJOR and fundamental check that author should provide answers:
- How are formed “pharmacy teams” in study, please describe. What that means?
(Pharmacy teams could comprise of pharmacists, pharmacy technicians, trainee pharmacists, trainee pharmacy technicians, dispensary staff and medicines counter assistants)
- Please add a calculation for sample size.
This was a service evaluation so a power calculation was not calculated beforehand
- How is performed initial analysis of data? Please specify.
We have amended in the methodology text
- The most important question is about ethical concerns. Why authors mentioned that there is no need for approval. In any case, with or without identifying the patients, authors must obtain the ethical consideration or consent at least (Please see declaration of Helsinki). This sentence: “Institutional ethical approval was also not required as patient identifying data were not collected.” Not true.
The text has been amended in the methodology to reference the NHS Regs tool and explained the manuscript is a service evaluation, not requiring ethical approval.
- Is the SnapSurvey licensed? Please add a number of license.
- From the results exclude the number of No response
This is a service evaluation and we included no response as this is important for the results to demonstrate the level of engagement
- Table 1 should be renamed; General findings are not appropriate. Please be more specific.
Findings from Patient response section of the Antibiotic Checklist
- All Figures must be labels, both X and Y axis must be with the explanation. Also, what present the bars in Figures, percents?? Please add in Figures.
- How you were sure in these data since there is type of self-reporting with no any controlling? How do you reduce the bias?
This is service evaluation providing a realistic prospective and we are not trying to control biases. We hope the study has high ecological validity due to the volume of data extracted.
- Do you have any previous pilot study to validate method tool? If you not, this results remain questionable as well as conclusions….
The details of this have been added to the Introduction
- In section Conclusion erase the implications, since they could be based e just on thinking not on the results. Only conclusions are based on results.
We have amended this so it explains this more
- The article is confused and very hard to read. There are a lot of abbreviations which reading keeps the more difficult, please reduce it.
We are following standard known abbreviations. The first time they appear in the manuscript, we have written them out in full. These are standard abbreviations commonly used in our field of work.
- Informed consent “All participants consented to participate and were aware of the study aims.” I am not sure about this. Please erase and obtain consent from the Institution to collect and use this data.
We hope that our response answers your queries. Thank you once again for your feedback.

Round 2
Reviewer 1 Report
Thank you for your response and addressing the aforementioned queries.
Reviewer 2 Report
The authors corrected all requested concerns. I suggest to accept.